# Multivalent 9-O-Acetylated-sialic acid glycoclusters as potent inhibitors for SARS-CoV-2 infection

Simon J. L. Petitjean[1,6], Wenzhang Chen[2,6], Melanie Koehler [1,6], Ravikumar Jimmidi[2], Jinsung Yang [1], Danahe Mohammed[1], Blinera Juniku[1], Megan L. Stanifer [3,4], Steeve Boulant[4,3], Stéphane P. Vincent [2✉] & David Alsteens [1,5✉]

The recent emergence of highly transmissible SARS-CoV-2 variants illustrates the urgent need to better understand the molecular details of the virus binding to its host cell and to develop anti-viral strategies. While many studies focused on the role of the angiotensin-converting enzyme 2 receptor in the infection, others suggest the important role of cell attachment factors such as glycans. Here, we use atomic force microscopy to study these early binding events with the focus on the role of sialic acids (SA). We show that SARS-CoV-2 binds specifically to 9-O-acetylated-SA with a moderate affinity, supporting its role as an attachment factor during virus landing to cell host surfaces. For therapeutic purposes and based on this finding, we have designed novel blocking molecules with various topologies and carrying a controlled number of SA residues, enhancing affinity through a multivalent effect. Inhibition assays show that the AcSA-derived glycoclusters are potent inhibitors of cell binding and infectivity, offering new perspectives in the treatment of SARS-CoV-2 infection.

[1] Louvain Institute of Biomolecular Science and Technology, Université catholique de Louvain, Louvain-la-Neuve, Belgium. [2] Laboratory of Bio-Organic Chemistry (NARILIS), UNamur, Namur, Belgium. [3] Dept. of Infectious Diseases, Medical Faculty, Center for Integrative Infectious Diseases Research (CIID), University of Heidelberg, 69120 Heidelberg, Germany. [4] Department of Molecular Genetics and Microbiology, College of Medicine, University of Florida, Gainesville, USA. [5] Walloon Excellence in Life sciences and Biotechnology (WELBIO), Wavre, Belgium. [6] These authors contributed equally: Simon J. L. Petitjean, Wenzhang Chen, Melanie Koehler. ✉email: stephane.vincent@unamur.be; david.alsteens@uclouvain.be

The recent outbreak of the severe acute respiratory syndrome coronavirus-2 (SARS-CoV-2) is a significant threat to human health and societies across the globe. Vaccination campaigns are flourishing all around the world and some light starts to appear at the end of the tunnel. However, at the same time, SARS-CoV-2 variants of concern, that escape neutralization by vaccine-induced immunity, are appearing[1]. In this context, there is an urgent need to obtain a full picture of the cellular factors, and entry pathways involved in SARS-CoV-2 infection to develop new broad-spectrum antiviral therapies.

SARS-CoV-2 gains its foothold to the host cell surface first via attachment factors (e.g. sialic acid[2], heparan sulfate[3], etc.) and then specific receptors (e.g. angiotensin-converting enzyme 2 (ACE2), neuropilin-1)[4,5] through interactions with its spike viral glycoprotein (S protein) (Fig. 1a). The S protein assembles into a homotrimer and comprises two functional subunits: the S1, responsible for receptor recognition and S2, that triggers membranes fusion (Fig. 1b)[6,7]. The S1-subunit itself can be divided into the N-terminal domain (NTD) containing the glycan-binding domain (GBD) and the C-terminal domain (CTD) accommodating the receptor-binding domain (RBD) (Fig. 1b). The GBD engages glycoproteins and glycolipids in most CoVs[8], whereas the RBD binds to the ACE2 receptor[9].

While S protein binding to ACE2 has been already extensively studied and a consensus reached about its central role in infection, several studies suggest a pivotal role of attachment factors that appear necessary for infecting ACE2-positive cells, such as heparan sulfate and sialic acids (SA)[10–12]. However, the exact role of sialoglycans binding by SARS-CoV-2 spike (S) protein remain currently unclear. Those interactions, even in the form of viral surfing, could be very important and determine the infection outcome[13,14]. Thus, targeting this step can be an efficient way to inhibit infection. Other sialylated glycolipids and/or glycoproteins exposed at the host cell surface may also be involved in weak, but selective, interactions with the virus. SA consists of nine-carbon sugar neuraminic acid derivatives frequently present at the non-reducing end of cell surface oligosaccharides on glycolipids and glycoproteins[15]. Composition and complexity of the chain, glycosidic linkage, acetylation, methylation and sulfation give rise to a huge diversity of SA present on our cells[16,17]. They are often used as primary and crucial factors by viruses causing lung infection, including many CoVs such as Middle-East respiratory syndrome (MERS)-CoV but not SARS-CoV[8,18]. A very recent study revealed that the S protein on SARS-CoV-2 recognizes oligosaccharide containing SAs, with a strong preference for

monosialylated gangliosides[12]. Another study supplied a first experimental demonstration of the presence of a α2,3- and α2,6-sialyl N-acetyllactosamine binding site in the NTD of the S1[19]. Here we will focus on 9-O-acetylated-SA (9-AcSA) (Fig. 1a, left), that are of particular interest, since it has been shown that coronaviruses OC43 and HKU bind this glycan via a conserved receptor-binding site in their spike protein[17]. In addition, cryo-EM and X-crystallography reveal the SA-binding site of MERS-CoV to be a groove-like sunken domain within the NTD (highlighted as gray circle in Fig. 1b)[20]. Comparison of SARS-CoV-2 S protein with the spike protein of other CoVs supports even more a role for 9-AcSA in SARS-CoV-2 infection. A first structure-based sequence comparison of the NTD highlights three loop regions in SARS-CoV-2 not present in SARS-CoV, but similar to the MERS-CoV SA-binding pockets[20]. This finding was corroborated by molecular docking simulations and electronic density mapping, predicting also the presence of a MERS-CoV-like SA-binding site in the NTD of the SARS-Cov-2[21]. A recent cryo-EM study also reveals similarities in the NTD with the bovine coronavirus (BCoV), suggesting a preferable binding for 9-AcSA[22]. Altogether, those evidence have led to the proposition of a flat and non-sunken SA-binding domain of 290 amino acid residues close to the RBD domain in the S1 subunit of the SARS-CoV-2 spike protein (highlighted as purple circle in Fig. 1b)[11]. However, other studies suggest that glycosylation of ACE2 receptor-bearing SA residues do not act as additional anchoring points, but rather hinder SARS-CoV-2 engagement[23–25]. In this context, microarray studies failed to evidence SA binding[26]. Therefore, there is a clear need to clarify the role of SA in SARS-CoV-2 infection.

In this work, we used force-distance (FD) curve-based atomic force microscopy (AFM)[27,28] to investigate the biophysical properties of purified S1-glycoprotein or SARS-CoV-2 virions binding to sialic acids, on both model surfaces and living cells. Focusing on the potential influence of acetylation in the binding process, we demonstrated SARS-CoV-2 specifically interacts with 9-AcSA and extracted the kinetic and thermodynamic properties of their bond formation, revealing a moderate affinity interaction, typical of most glycans-virus bonds. Consequently, this specific recognition makes them a competitive inhibitor for SARS-CoV-2 binding. Thus, we designed novel blocking molecules bearing multiple SA moieties to compensate their modest affinity and exploit the so-called multivalent effect[29]. We synthetized and tested four multivalent glycoclusters (with different topologies and valencies) presenting either SA or 9-AcSA. We identified

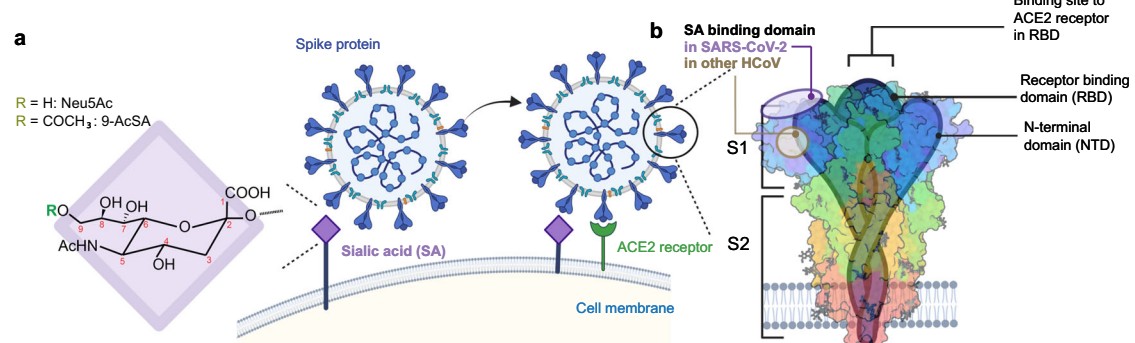

**Fig. 1 SARS-CoV-2 binding to cell surface receptors. a** SARS-CoV-2 spike protein interacts with both cell surface sialic acids and ACE2 for virus cell entry. The sialic acid family includes many derivatives of the nine-carbon sugar neuraminic acid. The *N*-acetyl-neuraminic acid (Neu5Ac) and 9-O-acetylated sialic acid (9-AcSA) are presented here. The S glycoprotein is composed of two subunits, S1 and S2, and is commonly represented as a sword-like spike. **b** Protein structure model of SARS-CoV-2 S protein showing the RBD binding domain for ACE2 receptor on top and the flat, non-sunken sialic acid-binding domain for host cell surface sialic acid-binding (purple circle) (PDB ID: 6VXX). The groove-like, sunken SA-binding domain in other human coronaviruses (HCoV) is highlighted in the brown circle. Created with BioRender.com.

9-AcSA-derived porphyrin having high-binding inhibitory capacity (in the sub-μM range) both on purified receptors and on living cells. Finally, using infection assays on living cells, we showed this molecule has a very promising neutralization potential.

## Results

**S1 subunit preferentially binds to 9-*O*-acetylated sialic acid**. To study the role of sialic acid during the first steps of SARS-CoV-2 binding to cell host surface, we used FD curve-based AFM[30–32] to compare SARS-CoV-2 binding to 9-AcSA and SA (*N*-acetylneuraminic acid (Neu5Ac)) and characterize the binding free-energy landscape of the interaction to 9-AcSA. To mimic exposure of cell-surface glycans in vitro, biotinylated sialic acids, either biot-9-AcSA or biot-SA (see synthesis in supplementary information), were immobilized onto streptavidin-coated surfaces[31] and validated by AFM imaging and scratching experiments, revealing a $1.1 \pm 0.1$ nm thick deposited layer (Supplementary Fig. 1a), which is in line with previously reported result[31]. The interaction between the spike S1 subunit and the sialic acid-coated surfaces was monitored by FD-based AFM (Fig. 2a). Individual FD curves were sorted based on the presence or not of a specific adhesion event (based on the rupture distance of the unbinding event and on the shape of the FD curve) providing access to the overall binding probability (BP) (Fig. 2b, c). Strikingly, we observed a ~3 fold-significant difference between acetylated and non-acetylated SA, suggesting a higher avidity for 9-AcSA. In addition, we confirmed the specificity of the interaction by conducting additional independent control experiments. Both competition assays with free 9-AcSA (Fig. 2c) or surface coated with only streptavidin (without SA) resulted in significantly lower BP, confirming the specificity of the 9-AcSA interaction with SARS-CoV-2 S1 domain.

**Exploring the dynamics of the interaction**. Next, we characterized the binding free-energy landscape of the interaction between S1 and 9-AcSA (Fig. 2d), using single-molecule dynamic force spectroscopy (DFS). Exploring the binding strength of the bond over a wide range of loading rates (i.e. the load applied on the bond over time) and fitting the single bond rupture force using the Bell-Evans model[33,34] enables to extract the kinetic off-rate ($k_{off}$) of the interaction and the distance to the transition state ($x_u$). To further promote single interactions, we decreased the density of 9-AcSA groups on the surface by co-incubating it with 25% free biotin (see methods) and we identified single versus multiple bond rupture events as previously established (Supplementary Fig. 2a)[35,36]. The mean forces of the single rupture events (larger black dots plotted over mean LR of this range in Fig. 2e) were fitted with the Bell-Evans model, leading to a $k_{off}$ value of $0.24 \pm 0.2 \, s^{-1}$ and $x_u$ of $0.5 \pm 0.2$ nm (Fig. 2e). These values are consistent with data reported for other virus-glycan interaction pairs[31,37].

Next, by monitoring the influence of contact time on BP (Fig. 2f), we were able to estimate the association rate ($k_{on}$) by assuming that the receptor-bond complex can be approximated by a pseudo-fist-order kinetics[36]. By fitting the data with a mono-exponential growth model (more details in methods), we extracted a $k_{on}$ of $(4.2 \pm 0.2) \times 10^4 \, M^{-1}s^{-1}$ and calculated the affinity constant ($K_D$) as the ratio between $k_{off}$ and $k_{on}$, resulting in a value of $5.7 \pm 5$ μM, several orders of magnitude higher than the previously measured affinity for SARS-CoV-2 towards gangliosides[12]. Values in the μM range correspond to a moderate affinity, supporting the hypothesis that SARS-CoV-2 uses glycans as a first moderate-affinity foothold on the host cell surface, facilitating subsequent strong binding to ACE2 receptor[18].

The values reported here support even more the key role of acetylation in SA recognition by SARS-Cov-2.

**Binding at the SARS-CoV-2 virion level**. To evaluate the physiological relevance of the probed interaction, we used non-replicating SARS-CoV-2 particles, i.e. native SARS-CoV-2 virions inactivated through UV radiation, leading to photochemical damage to nucleic acids while maintaining particle integrity (Supplementary Fig. 1b), and subsequently to the disruption of viral replication[38]. We evaluated the binding of the non-replicating SARS-CoV-2 particles to 9-AcSA, by grafting the whole virions onto the AFM tip (Fig. 2g). We probed the interaction at moderate (1 μm s$^{-1}$) and fast (20 μm s$^{-1}$) pulling speed (Fig. 2h, small red and blue dots, respectively), and reconstructed the DFS plot that we overlaid with the data obtained with the purified S1 domain (Fig. 2i and Supplementary Fig. 2b). We observed good agreement between the data collected with the purified S1 domain only or with the full virions. Both the different rupture forces at the single-molecule level are very close but also the kinetic parameters extracted via the Bell-Evans model. At the virion level, higher forces compatible with multiple contacts were also recorded and attributed to multivalent interactions that are loaded and ruptured in parallel as confirmed by the predictive Williams-Evans fits (Fig. 2i, dashed lines)[39].

**Validation of the interaction on living cells**. Next, we validated the interaction directly on living cells by probing the interaction between either purified S1 or full non-replicative virions and a co-culture of unlabeled CHO, naturally expressing SA, and Lec2 cells (fluorescently labeled with a nuclear protein H2B-GFP and actin-mCherry). Although the sialylation pattern of CHO cells is not fully elucidated, it has been shown to support the infection by bovine coronavirus that use the 9-AcSA as entry receptor[40], while the Lec2 is a mutant cell line derived from the parental CHO cell line with a 70-90% deficiency in SA expression[41]. Furthermore, both cell lines do not express ACE2 receptors, making them an ideal choice for assessing SA binding[42]. To evidence the role of SA as surface receptors, we performed laser-scanning confocal microscopy on a co-culture of CHO and Lec2 cells and observed that UV-inactivated SARS-CoV-2 virions (fluorescently labeled with Atto488-NHS dye) preferentially bind to SA-expressing CHO cells (Supplementary Fig. 3). Then, using AFM tips functionalized with either S1 glycoprotein or the full virions, we scanned by AFM a confluent monolayer of co-cultured CHO and Lec2 cells using conditions to propagate both cell types (Fig. 3a, e)[32,35]. Guided by fluorescence, we chose areas of view in which both cell types were adjacent, serving as a direct internal control in AFM imaging[31]. In such areas, we simultaneously recorded AFM height images (Fig. 3b, f) together with their corresponding adhesion maps, revealing the location of specific adhesion events displayed as bright pixels on the adhesion map (Fig. 3c, g). As anticipated, CHO cells showed a significantly higher density of adhesion events (~9% for S1, ~13% for SARS-CoV-2, Fig. 3d, h, $N = 10$ cells for S1 and $N = 12$ cells for SARS-CoV-2), whereas Lec2 cells displayed only a sparse distribution of these events (~3% for S1, ~5% for SARS-CoV-2, Fig. 3d, h, $N = 10$ cells for S1 and $N = 12$ cells for SARS-CoV-2), confirming the establishment of specific SARS-CoV-2 bonds to SA on living cells. In addition, we extracted from the individual FD curves the specific binding forces (and corresponding LR) following the same strategy as for the model surfaces (Fig. 3i, j) and observed that the DFS data recorded with both S1 and full virions on the living cells are in good agreement with the data previously obtained on purified AcSA (Fig. 3k and Supplementary Fig. 4). Probing the interaction in a cellular context brings physiological relevance to this work as SA is displayed on

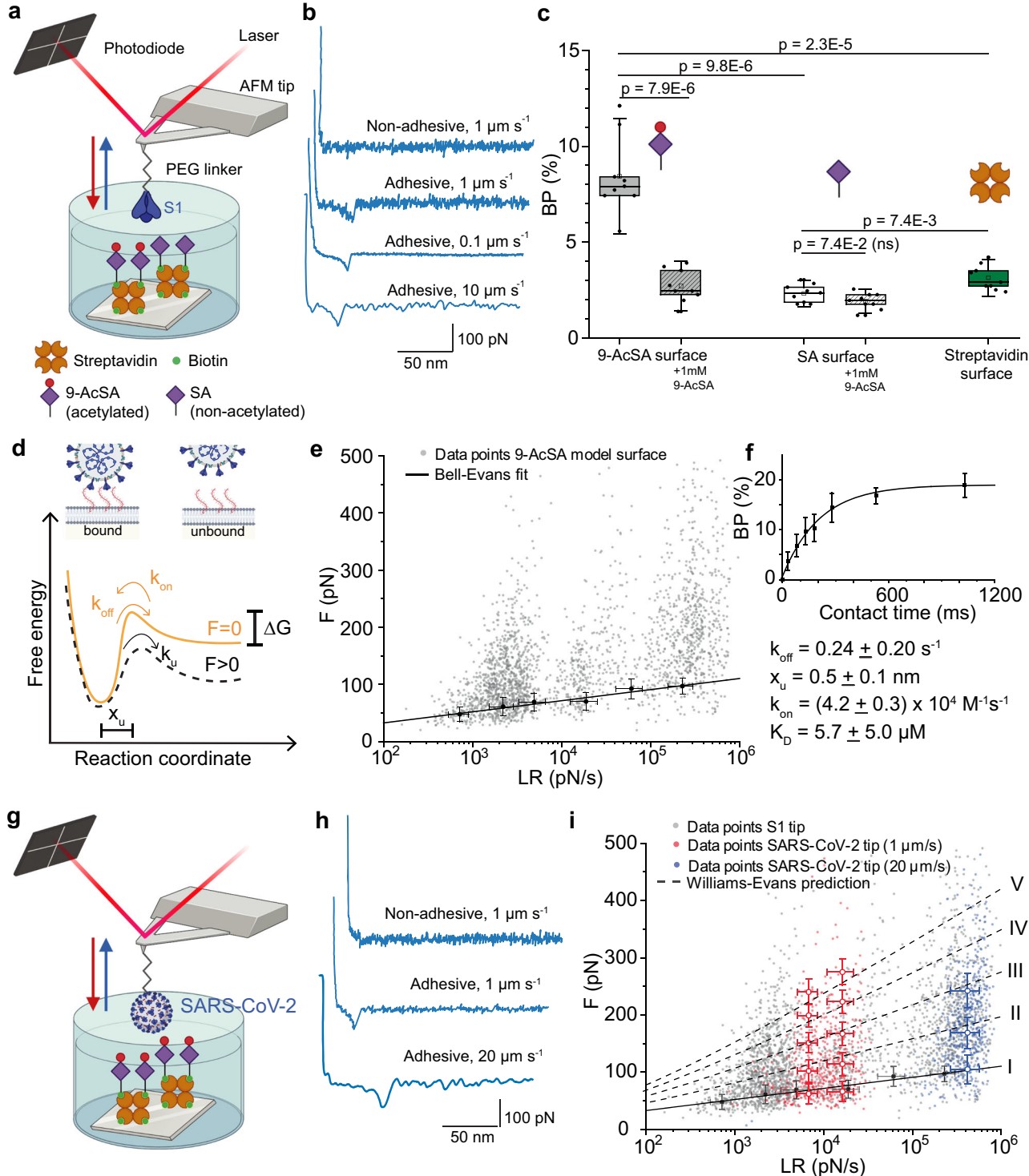

various glycoconjugates among a dense meshwork of different glycans composing the glycocalyx on the cellular surface. The significant difference in BP between CHO and Lec2 cells as well as the good alignment of the data on the DFS plot confirm the establishment of specific interactions between SARS-CoV-2 virions and SA in a complex cellular environment.

**Synthesis of multivalent SA and 9-AcSA glycoclusters.** After confirming and characterizing the interaction between SARS-CoV-2 and SA, we synthesized several multivalent glycoclusters

presenting sialic acids as potential competitors of this interaction. The designed glycoclusters hold two key features: (i) a systematic comparison of the effect of 9-*O*-acetylation on the sialic acid ligands and (ii) a multivalent display using different central scaffolds of distinct valencies and topologies. Pillar[5]arene (tubular and decameric), calix[4]arene (conic and tetrameric), fullerene (globular and dodecameric), and porphyrin (planar and tetrameric) were selected as central scaffolds for their distinct and complementary ways to spatially distribute their ligands, a key parameter to achieve efficient enhancement of binding affinities with receptors[43,44] (Fig. 4). To build the targeted glycoclusters,

**Fig. 2 Evaluating SARS-CoV-2 binding to SA and extraction of their kinetic and thermodynamic properties. a** AFM tip derivatized with the S1 domain of SARS-CoV-2 is probed against either 9-AcSA or SA coated surface, immobilized by streptavidin-biotin binding. **b** Representative retraction parts of FD curves recorded at different pulling speeds, showing either no adhesion or specific adhesion events. **c** Box plot of specific binding probabilities (BP) measured between the S1 functionalized tip and the surface coated with 9-AcSA, SA, or streptavidin before and after blocking with free 1 mM 9-AcSA. One data point represents the BF obtained for 1024 FD curves. The square in the box indicates the mean, the colored box the 25th and 75th percentiles, and the whiskers the s.d. of the mean value. The line in the box indicates the median. $N = 9$ maps examined over three independent experiments. $P$-values were determined by two-sample $t$-test in Origin. **d** Bell-Evans model describing a virus–SA bond as a two-state model. The bound state is separated from the unbound state by a single energy barrier located at a distance $x_u$. $k_{off}$ and $k_{on}$ represent the dissociation and association rate constants, respectively. **e** Dynamic force spectroscopy (DFS) plot showing the distribution of the rupture forces as a function of their LR (grey dots, each corresponding to a single FD curve) and average rupture forces (larger black dots), determined at six distinct loading rate (LR) ranges measured between 9-AcSA and S1 ($N = 2331$ data points). Data corresponding to single interactions were fitted with the Bell-Evans (BE) model (straight line). **f** The binding probability (BP) is plotted as a function of the hold time. Least-squares fits of the data to a mono-exponential decay curve (line) provides average kinetic on-rates ($k_{on}$) of the probed interaction. Further calculation ($k_{off}/k_{on}$) leads to $K_D$. One data point represent the mean BF obtained for $N = 3$ independent experiments representing 1024 FD curves each. **g** Probing binding of UV-inactivated SARS-CoV-2 to 9-AcSA model surface. **h** Representative retraction curves. **i** DFS plot showing the distribution of the rupture forces as a function of their LR (small dots, each corresponding to a single FD curve) and average rupture forces (larger dots) measured either between S1 subunit and 9-AcSA (taken from **e**), or between SARS-CoV-2 and 9-AcSA at 1 μm s$^{-1}$ ($N = 512$ data points, red) or 20 μm s$^{-1}$ ($N = 504$ data points, blue). Solid line and dashed lines correspond to the Bell-Evans (BE) and Williams-Evans (WE) model, respectively. The numerals I, II, III, and IV stand for the single, double, triple, and quadruple interaction, respectively. All experiments were reproduced at least three times with independent tips and samples. The error bars indicate s.d. of the mean value. Cartoons in panels **a**, **d**, **g** were created with BioRender.com. Source data in panels **c**, **e**, **f**, **i** are provided as a Source Data file.

two clickable sialic acid derivatives **1** (9-OH) and **2** (9-*O*-Ac) were prepared first (Fig. 4) and then clicked to the four central scaffolds mentioned above. Thanks to a protocol allowing the selective acetylation of a primary alcohol in presence of secondary alcohols[45], alkyne **2** was obtained from known glycoside **1**[46,47] in 39% yield. The regioselective acetylation of 9-OH on **1** was confirmed by the shift of proton H-9 and carbon C-9 to the lower field in the $^1$H and $^{13}$C NMR spectra, respectively, and by $^1$H-$^1$H COSY and $^1$H-$^{13}$C HMQC experiments. The same shifts were observed in the final glycoclusters **5**, **8**, **11**, and **14** (see Supplementary Note 1).

Then, the clickable α-propargyl sialic acids **1** and **2** were grafted to multimeric azides **3**[48], **6**[49], **9**[50], and **12**[51] (structures drawn in Supplementary Note 1), using either a combination of copper(II)sulfate and sodium L-ascorbate, or copper(I)bromide dimethyl sulfide[49]. More specifically, an excess amount of **1** or **2** (12 eq.) was coupled to tetra-azide pillar[5]arene **3** with a catalytic amount of copper(II)sulfate (2 eq.), and sodium L-ascorbate (6.6 eq.) in a mixed solvent system (1,4-dioxane/H$_2$O, 2:1) at room temperature overnight. Then, **4** and **5** were precipitated with acetone and purified by copper scavenging (Quadrasil MP) and size-exclusion Sephadex®G-25 chromatography. The purified decavalent pillar[5]arenes **4** and **5** were obtained in 74% and 62% yield, respectively. A specific method was optimized for the porphyrin tetramers **10** and **11**, to avoid ion exchange between copper and zinc and to cope with the solubility properties of **9** (ref. [52]). Thus, a lower catalyst amount (0.4 eq.) and a ternary solvent system (THF/DMSO/H$_2$O, 3:3:1) were employed. The tetravalent porphyrin conjugates **10** and **11** were obtained in 61% and 87% yields, respectively. The calix[4]arenes **7** and **8** and fullerenes **13** and **14** were also obtained in high yields using similar coupling and purification protocols. All the multimeric species were characterized by $^1$H, $^{13}$C, and mass spectrometry (see Supplementary Note 1) to ascertain the completion of all cycloadditions.

**Inhibition of SARS-CoV-2 binding using SA-derived glycoclusters**. The eight glycoclusters either decorated with the SA or 9-AcSA were tested for their blocking properties at increasing concentrations (from 0 to 100 μM) using our SMFS approach and full non-replicative SARS-CoV-2 particles (Fig. 5a). Inhibition assays were performed with full virions to account for the multivalence that is key for that kind of moderate affinity biological interaction. First, we evaluated the blocking capacity of the monovalent, commercially available 9-AcSA (Carbosynth) at 0, 1, 10, and 100 μM (Fig. 5b). We observed only a slight reduction of 20-30% in the binding probability over the explored concentration range. This low efficiency was expected as we determined earlier that AcSA displays a moderate affinity for the SARS-CoV-2 spike. Therefore, a multivalent display is commonly required to compensate this disadvantage by exploiting the so-called multivalent effect observed in many natural and artificial systems to enhance the affinity/avidity of a ligand towards its receptor. Thus, we compared the four glycoclusters either functionalized with SA or 9-AcSA. The relative BP plots (Fig. 5c–f) show either no or slight inhibition for the SA-clusters, whereas a progressive and more effective reduction in the BF is observed for all 9-AcSA-clusters. The absence of specific inhibition for the SA-glycoclusters was expected as we determined earlier that the interaction requires the acetylation to occur (see Fig. 2c). Hence, it accommodates for a successful control of the inhibition caused by the 9-AcSA-clusters. For the latter, we observed for all 9-AcSA-clusters a 50% reduction around 10 μM. The 9-AcSA porphyrin tetramer **11** seems the most efficient inhibitor reaching a ~50% reduction already at 1 μM, suggesting a 50% inhibitory concentration (IC$_{50}$) in the same range (Fig. 5e, g).

**9-AcSA-porphyrin tetramer 11 decreases SARS-CoV-2 infectivity on cells**. As our screening pointed out the porphyrin **11** as the most efficient 9-AcSA-derived oligomer for its anti-binding properties, we characterized it in more detail, i.e. at lower concentrations and on living cells. Stunningly, we found that the 9-AcSA porphyrin tetramer **11** leads to a reduction of >50% of the probed SARS-CoV-2 – 9-AcSA interactions already at 10 nM (Fig. 6a) and follows an exponential decay of as a function of the concentration. Next, we probed the inhibition potential on living cells and confirmed this trend also under physiological relevant condition (Fig. 6b and Supplementary Fig. 5).

Then, beyond the anti-adhesive capacities, we wanted to evaluate the neutralization properties of the 9-AcSA-porphyrin tetramer **11**, i.e. its capacity to prevent the entry of the virus and thus the infection. In order to evaluate its neutralization potential, we used a robust virus infectivity assay, which can be used at low biosafety conditions[53]. Briefly, propagation-incompetent G-deleted vesicular stomatitis virus (VSV) trans-complemented with the SARS-CoV-2 spike protein and encoding for a GFP

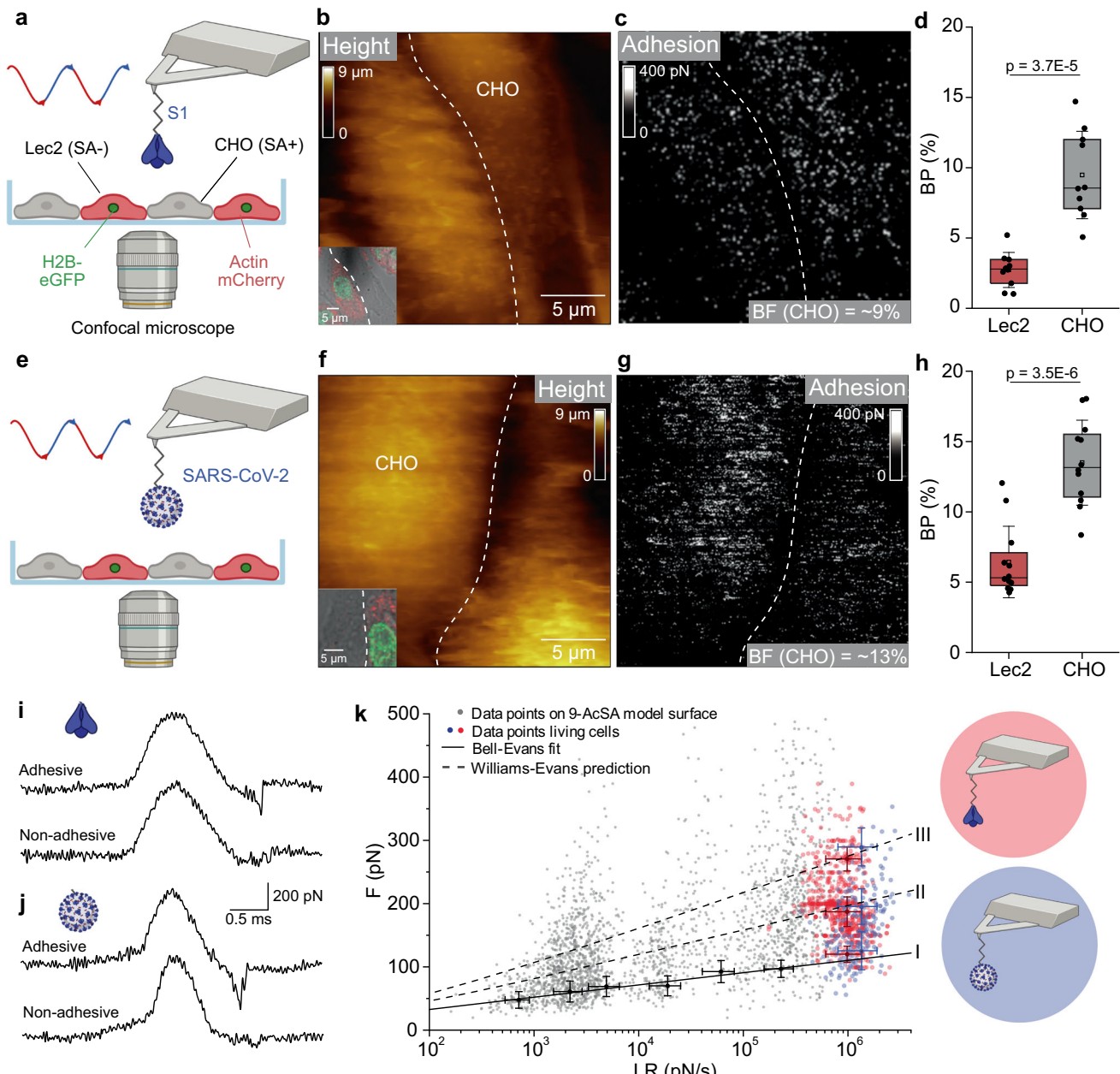

**Fig. 3 Probing SARS-CoV-2 binding to SA on living cells. a, e** Combined confocal microscopy and FD-based AFM assay of S1 (**a**) or SARS-CoV-2 (**e**) binding to cells expressing (CHO) or lacking (Lec2) SA on the cell surface. **b, c, f, g** FD-based AFM topography image (**b, f**) and corresponding adhesion map (**c, g**) from probing of adjacent cells (the inserts in **b, f** show the optical images of the probed area) with either S1 (**b, c**) or SARS-CoV-2 (**f, g**) functionalized AFM tips. Both adhesion maps show mostly interactions on CHO cells (SA-expressing cells) (white pixels). **d, h** Box plot of the BP between S1 and Lec2/ CHO cells (**d**) or SARS-CoV-2 and Lec2/ CHO cells (**h**). The square in the box indicates the mean, the colored box the 25th and 75th percentiles, and the whiskers the s.d. of the mean value. The line in the box indicates the median. $N = 10$ (**d**) or 12 (**h**) maps examined over three independent experiments. *P* values were determined by two-sample *t* test in Origin. **i, j** Force versus time curves showing either a non-adhesive curve (bottom) or specific adhesive curves acquired for CHO and S1 (**i**) or SARS-CoV-2 (**j**) interaction. **k** DFS plot of data from acetylated SA model surfaces (grey circles, from Fig. 2e) and living cells (red dots: S1-tip, $N = 655$; blue dots: SARS-CoV-2-tip, $N = 200$). Data are representative of at least $N = 12$ cells from $N = 4$ independent experiments. Cartoons in panels **a, e, i–k** were created with BioRender.com. Source data in panels **d, h, k** are provided as a Source Data file.

reporter protein (VSV-SARS-CoV-2) was used on A549 cells and A549 cells transduced with the ACE2 receptor with or without interfering molecules at various concentrations (Fig. 6c–e and Supplementary Fig. 6). A549-ACE2 were infected with a MOI of 5 of the VSV-SARS-CoV-2. Infectivity was monitored by measuring the GFP fluorescence in the cells 24 h post-infection (Fig. 6c–e and Supplementary Fig. 6). While A549 cells are not infected by the VSV-SARS-CoV-2 (Fig. 6d, 1st line), overexpression of

ACE2 strongly enhanced infection (Fig. 6d, 1st line). Next, we performed the infectivity assay while incubating the VSV-SARS-CoV-2 with SA, 9-Ac-SA, SA-porphyrin **10,** and 9-AcSA-porphyrin **11.** While both monovalent sialic acids do not significantly reduced VSV-SARS-CoV-2 infectivity, we observed that the 9-AcSA-porphyrin significantly reduced VSV-SARS-CoV-2 infectivity with an estimated IC50 in the range 0.1–1 µM (Fig. 6e). This cell-based assay confirms our previous results on

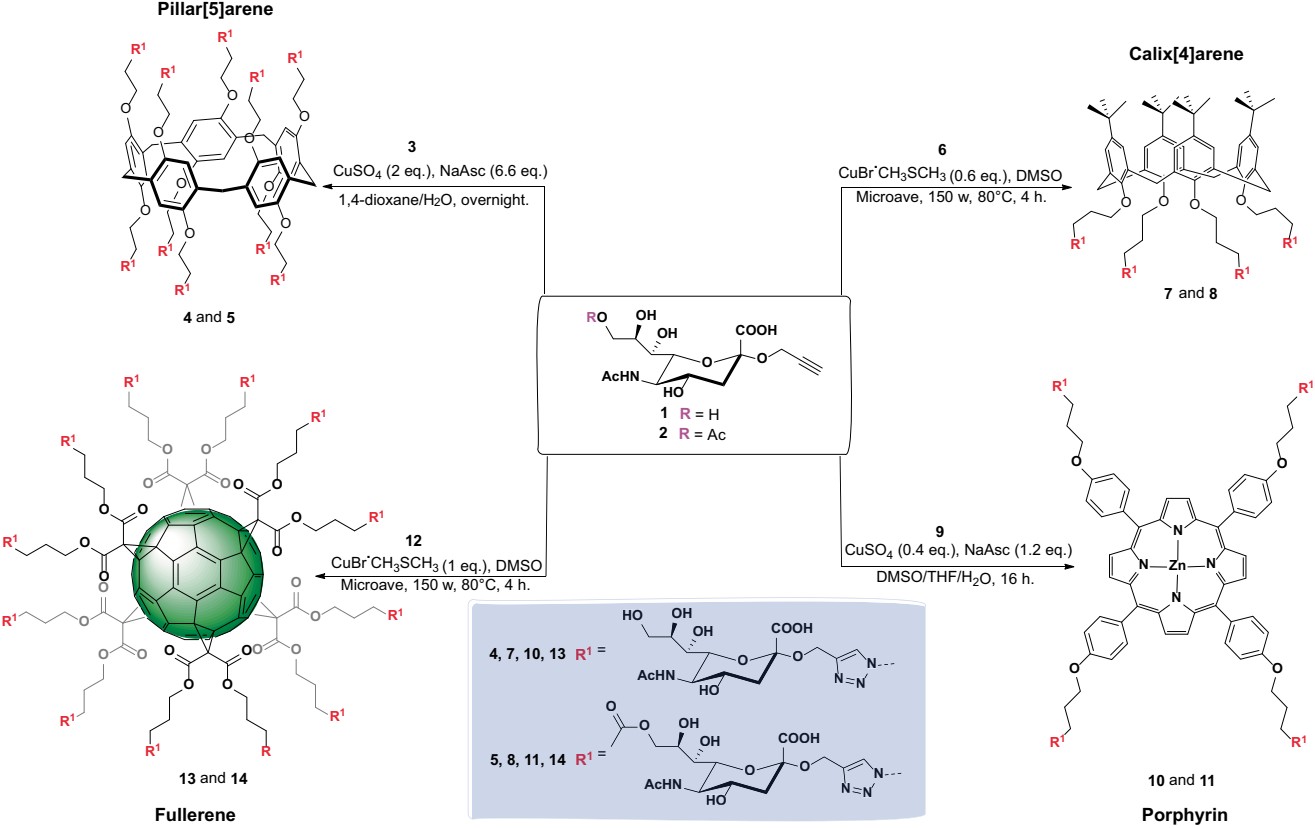

**Fig. 4 Synthesis of the targeted multimeric sialic acid glycoclusters.** 2-*O*-Propargylated sialic acids **1** and **2** were first prepared from sialic acid (see SI) in 5 and 6 steps, respectively. Then, **1** and **2** were clicked to the four azidated oligomers using coupling protocols specifically optimized for each scaffold (Yields: **4** (74%), **5** (62%), **7** (92%); **8** (81%); **10** (61%); **11** (87%), **13** (81%), and **14** (82%)).

the effect of sialic acids on SARS-CoV-2 binding and suggests that effective inhibition of virus binding to its receptor can lead to a significant drop in infectivity.

## Discussion

AcSAs have been identified as important players in numerous human biological functions, including among others protection of the cell surface and glycoproteins from proteases[54,55], formation of the blood vessel lumen[56] or immune cell trafficking[57]. Besides homeostatic cellular function, AcSA are usurped as binding sites for several human pathogens, including viruses. One of the first entry points on the respiratory virus pathway is the airway epithelium, rich in 7/9-AcSA, that are therefore used as attachment sites by many respiratory viruses, such as influenza (C and D types[58,59]) and coronaviruses (HCoV-OC43 and HCoV-HKU1[8,60]). As cryo-EM has revealed a well-conserved binding pocket in many coronaviruses that engages 9-AcSA[11], we sought to prove that SARS-CoV-2 could share the same binding properties towards this sialylated glycan. Using FD-based AFM, we compared the binding properties of SARS-CoV-2 to SA and 9-AcSA. Specific binding was difficult to assess in the case of SA, as we only observed binding with a low probability which is directly related to a very low affinity, as it has been already evidenced by previous studies showing affinities in the millimolar range[31,37]. In comparison for binding to 9-AcSA, we observed clear binding and were able to extract a $K_d$ that is several orders of magnitude higher, in the low micromolar range (~μM) (Fig. 2). Our result provides a molecular framework suggesting that SARS-CoV-2 binds preferentially to 9-AcSA-decorated oligosaccharides present at the surface of host cells.

This weak affinity is counterbalanced by the oligomeric nature of the spike glycoproteins present at the viral surface that undoubtedly enhance adsorption to target receptors through avidity[61]. This was confirmed by our AFM experiments at the full-virions level, which revealed similar kinetic binding properties at the single bond level and also evidenced the formation of multivalent interactions.

A growing body of evidence identifies 9-O-AcSA as a common receptor for betacoronaviruses, indicating a convergent evolutionary adaptation to the human airway sialoglycoma. In this context, targeting this highly conserved attachment step, especially in the context where many SARS-CoV-2 variants are emerging, appears to be a therapeutic pathway with high potential. Based on our findings and the proven evidence for a specific binding, we have designed and evaluated new antiviral agents based on glycoclusters with variable geometry and valency decorated with 9-AcSA moieties for which we evidenced a specific binding. While monovalent 9-AcSA shows a high IC$_{50}$ > 100 μM, we observed for all 9-AcSA-derived glycoclusters screened an IC$_{50}$ in the range 1-10 μM confirming a multivalent effect for these molecules (Fig. 5). Among these, 9-AcSA-porphyrin even shows an IC$_{50}$ in the sub-micromolar range, both on purified 9-AcSA and in the cellular context, making it an excellent candidate as a therapeutic agent (Fig. 6a). This behavior can be explained by aggregation. Indeed, porphyrins, including glycosylated ones, can aggregate in solution. For instance, it was shown that the aggregative properties of a porphyrin bearing 4 iminosugars contributed to the multivalent inhibition of a mannosidase[62]. Using VSV-SARS-CoV-2 spike encoding for a GFP reporter, we next evaluated the neutralization potential of the 9-AcSA-porphyrin tetramer and observed a reduction of 50%

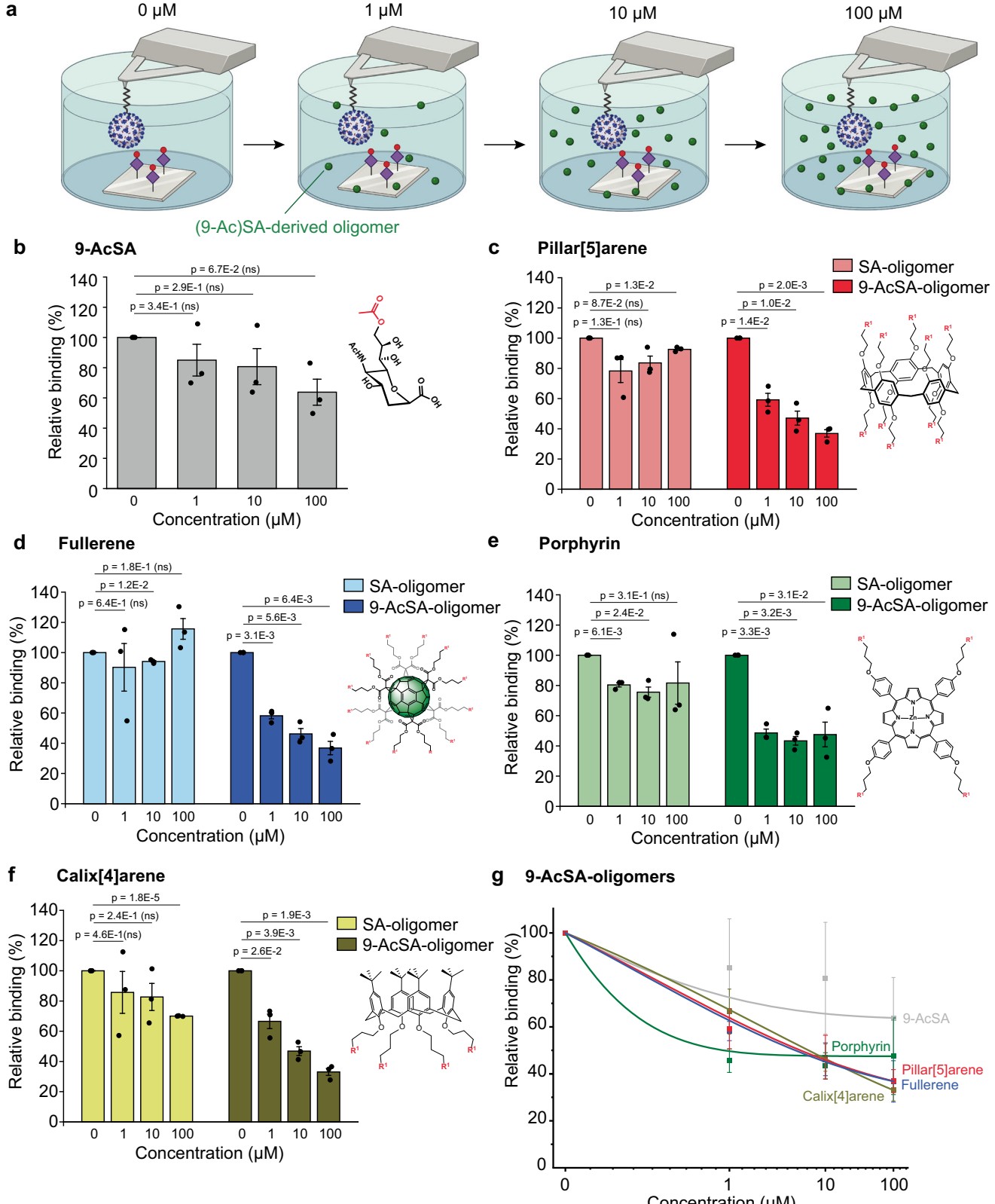

**Fig. 5 Screening of the anti-binding properties of SA-derived glycoclusters. a** The efficiency of blocking glycoclusters is evaluated by measuring the BP of the interaction between 9-AcSA coated surfaces and SARS-CoV-2 before and after incubation with the oligomers at increasing concentration (1–100 μM). **b**–**f** Normalized histograms showing the relative BP of the interaction between 9-AcSA and SARS-CoV-2 before and after incubation with 1, 10, or 100 μM of free 9-AcSA (**b**), Pillar[5]arene (**c**), fullerene (**d**), porphyrin (**e**), or calix[4]arene (**f**). **g** Graph showing the reduction of the BP. Data are representative of at least $N = 3$ independent experiments (tips and sample) per SA dendrimer concentration. *P*-values were determined by two-sample t-test in Origin. The error bars indicate s.d. of the mean value. Panel **a** was created with BioRender.com. Source data are provided as a Source Data file.

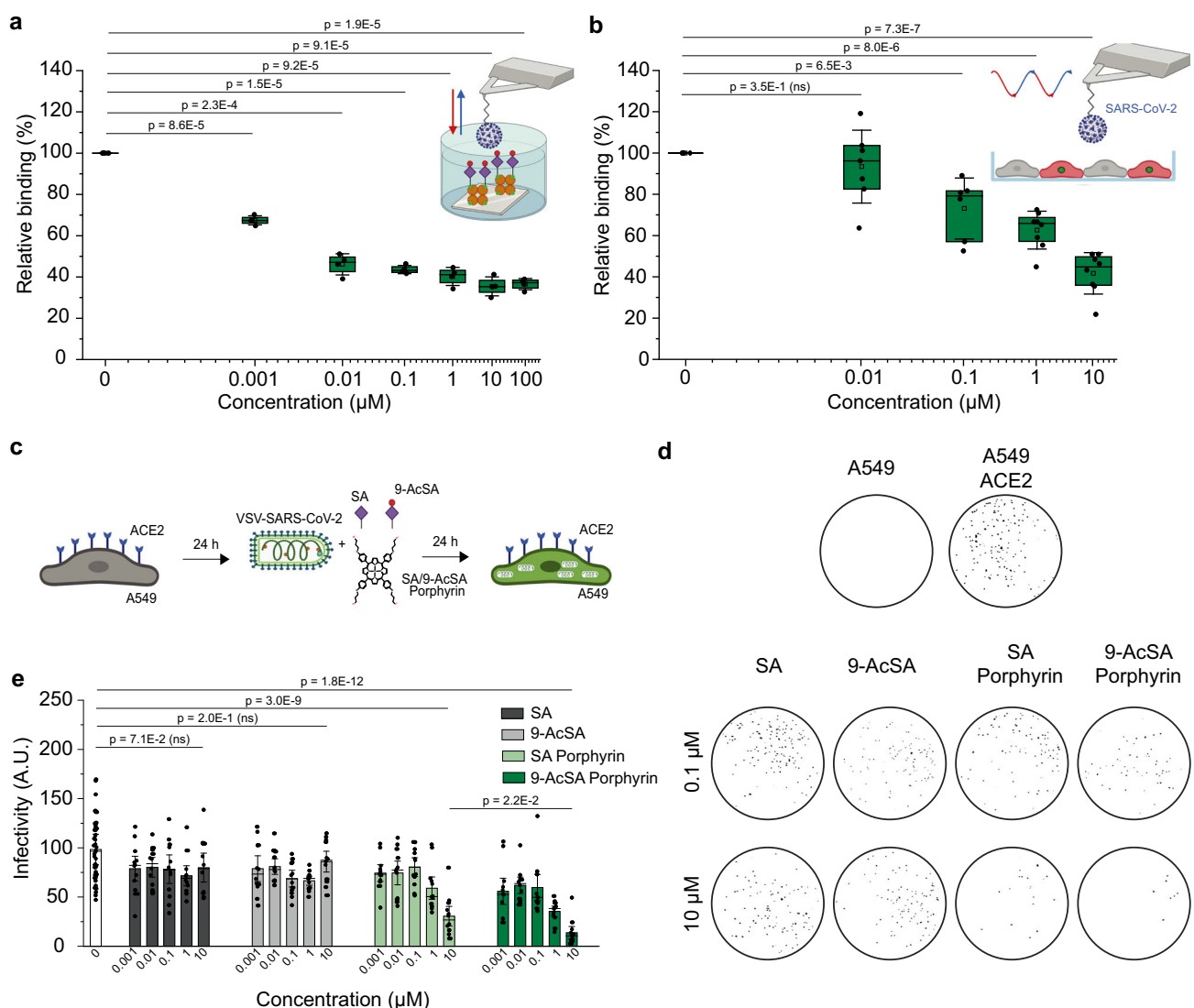

**Fig. 6 Characterization of AcSA-derived porphyrin 11 as a binding and infectivity inhibitor. a, b** Probing 9-AcSA-porphyrin efficiency to inhibit SARS-CoV-2 binding to acetylated SA on model surfaces and living cells at low concentrations. Box plot showing the relative binding values of the interaction between SARS-CoV-2 and **a** 9-AcSA model surfaces or **b** CHO cells before and after incubation with 9-AcSA-derived porphyrin 11 at increasing μM concentrations. The square in the box indicates the mean, the colored box the 25th and 75th percentiles, and the whiskers the s.d. of the mean value. The line in the box indicates the median. *P*-values were determined by two-sample *t* test in Origin. Data are representative of at least *N* = 3 independent experiments (tips and sample) per SA dendrimer concentration. **c–e** Infectivity assay. **c** Schematics: A549 and A549-ACE2 cells were seeded in 96-well plates for 24 h. SARS-CoV-2 spike pseudotyped virus and either SA, 9-AcSA, SA/9-AcSA porphyrin were incubated for 15 min at room temperature and added to the 96-well plate. The infectivity was measured 24 h later. **d** The images of infected cells in a well in the presence of either 0.1 μM or 10 μM of SA, 9-AcSA, SA/9-AcSA Porphyrins **10** and **11**. **e** The infectivity measured in the presence of SA, 9-AcSA, SA/9-AcSA porphyrin. Each dot shows the infectivity from a well. The colored box indicates the mean and the whiskers the s.d. of the mean value. The line in the box indicates the median. *P*-values were determined by two-sample *t* test in Origin. Cartoons in panels **a–c** were created with BioRender.com. Source data are provided as a Source Data file.

of the infectivity in the sub-μM range, demonstrating its ability to efficiently block SARS-CoV-2 infection. The aforementioned results demonstrate the important role of 9-AcSA in the interaction between SARS-CoV-2 and human airway epithelia and show that multivalent 9-AcSA-based inhibitors form a promising template for the design of antiviral drugs.

## Methods
**Generation of UV-inactivated SARS-CoV-2**. SARS-CoV-2 (BavPat1 strain, European Virology Archives) was grown in Vero E6 as previously described[63] and used at a passage 3. In all, 500 μL of passage 3 stock (supernatants of infected cells) was added to a six-well dish. The six-well dish was placed in a UV Stratalinker 1800 (Stratagene) without a lid and virus containing supernatant was exposed to 5000J UV irradiation. Virus inactivation was confirmed by adding 10 μL of the UV

treated supernatant to a 48-well plate containing 50,000 naïve Vero E6 cells and monitoring infection 48 h later by indirect immunofluorescence assay.

**Sialic acid-coated surfaces**. Gold-coated silicon substrates were first washed with ethanol and cleaned by UV-O treatment (Jetlight) for 15 min. The surfaces were then incubated overnight at 4 °C in a biotinylated bovine serum albumin (BBSA) solution (25 μg mL$^{-1}$ in PBS, Sigma). After rinsing with PBS, a drop of streptavidin (10 μg mL$^{-1}$ in PBS, Sigma) was pipetted onto the BBSA surface for 1 h at 4 °C, followed by rinsing with PBS. Finally, the BBSA-streptavidin surfaces were immersed for 1 h at 4 °C in a biotinylated biot-9-AcSA or biot-SA solution (10 μg mL$^{-1}$ in PBS; see Supplementary Note 2), followed by a final PBS rinsing. For DFS and $k_{on}$ experiments, the biotinylated sialic acids were co-incubated on the BBSA-streptavidin surfaces with 25% free biotin to ensure single-molecule interactions. The surfaces showed a homogeneous and stable morphology under repeated scanning and exhibited a thickness of 1.1 ± 0.1 nm (Supplementary Fig. 1a). The thickness of the deposited layer was estimated by scanning a small

area (1 μm × 1 μm) of the surface at high forces to remove the attached biomolecules, followed by imaging larger squares of the same region (5 μm × 5 μm) at a lower force.

**Virus particle imaging**. An 80 μL droplet of virus solution (~$10^7$ particles mL$^{-1}$) was deposited on a freshly cleaved mica substrate and incubated at +4 °C for 1 h. After rinsing 10 times with milliQ water, the sample was dried for 1 h at 37 °C. AFM imaging was conducted in the PeakForce Tapping mode using PeakForce-Hirs-F-A tips (nominal spring constant 0.4 N m$^{-1}$, Bruker) in air. The imaging parameters that were used are: tip oscillation frequency of 1 kHz, maximum peak force of 250 pN; scan rate of 0.25 kHz and displaying 256 pixels per line.

**AFM tip functionalization**. For AFM tip functionalization, NHS-PEG$_{24}$-Ph-alde-hyde linkers (Broadpharm) were used. AFM tips (MSCT-D probes, Bruker) were immersed in chloroform for 10 min, rinsed with ethanol, dried in a gentle stream of filtered nitrogen, cleaned for 15 min in an ultraviolet radiation and ozone cleaner (JetLight), and immersed overnight in an ethanolamine solution [3.3 g of ethanolamine hydrochloride in 6.6 mL of dimethyl sulfoxide (DMSO)]. The cantilevers were then washed three times with DMSO and three times with ethanol, and dried with nitrogen. Meanwhile, 3.3 mg of NHS-PEG$_{24}$-Ph-aldehyde linkers were dissolved in 0.5 mL of chloroform. The ethanolamine-coated cantilevers were immersed this solution together with 30 μL triethylamine. After 2 h incubation time, were washed three times with chloroform, dried with nitrogen, and placed in a star conformation (with the tips facing each other) on Parafilm (Bemis NA). In all, 50 μL of S1-subunit protein solution (0.1 mg mL$^{-1}$, Genscript Z03501) or UV-inactivated SARS-CoV-2 virions ($10^8$ particles mL$^{-1}$) and 2 μL of freshly prepared NaCNBH$_3$ solution (6 wt% vol-1 in 0.1 M NaOH(aq)) was pipetted on them and incubated for 1 h at 4 °C. Finally, 5 μL of 1 M ethanolamine (pH = 8) was added to the drop for 10 min to quench the reaction. After a wash in PBS, the tips were stored in PBS until the experiment.

**Binding probability assays and screening of inhibitory potential**. A Nanoscope Multimode 8 (Bruker) was operated in force-volume (contact) mode to conduct the force spectroscopy experiments on model surfaces (Nanoscope software v9.1). MSCT-D probes (nominal spring constant of 0.03 N m$^{-1}$) were used to record 5 μm × 5 μm maps, with a ramp size of 200 nm, a maximum force of 500 pN, and no surface delay. The sample was scanned using a line frequency of 1 Hz, and 32 pixels per line (32 lines). Both approach and retraction speed were kept constant at 1 μm s$^{-1}$.

To study the inhibitory potential of the synthesized 9-AcSA-derived glycoclusters on the binding affinity between SARS-CoV-2 and 9-Ac-SA, binding probabilities (fraction of curves showing binding events) were measured before and after incubation with different concentrations (0, 1, 10, and 100 μM respectively; additionally, at 1, 10, and 100 nM for porphyrin **11**) of the four SA-/ 9-AcSA-oligomers. Briefly, three force-volume maps were recorded on three different areas as described previously in the absence of any oligomer. Thereafter, the oligomer was added to the fluid cell and three maps were recorded for each concentration.

**Dynamic force spectroscopy**. Dynamic force spectroscopy experiments were performed using a ForceRobot 300 (JPK). Using the same parameters as for the binding probability assays, with a varying retraction velocity of 0.1, 0.2, 1, 5, 10, and 20 μm s$^{-1}$. Origin software (OriginLab) was used to display the results in DFS plots, and to generate rupture force histograms for distinct LR ranges and to apply various force spectroscopy models, as described elsewhere[35]. These models are used to quantify the energy landscape of this interaction and to extract the kinetic off rate $k_{off}$, as well as the distance to the transition state $x_u$.

For kinetic on-rate analysis, the BP was determined at a certain contact time ($t$) (the time the tip is in contact with the surface). Those data were fitted and $K_D$ calculated as described previously[36]. In brief, the relationship between interaction time ($\tau$) and BP is described by the following equation:

$$BP = A \times \left[1 - \exp\left(\frac{-(t - t0)}{t}\right)\right] \quad (1)$$

where $A$ is the maximum BP and $t_0$ the lag time. Origin software is used to fit the data and extract $\tau$. In the next step, $k_{on}$ was calculated by the following equation, with $r_{eff}$ the radius of the sphere, $n_b$ the number of binding partners, and $N_A$ the Avogadro constant

$$k_{on} = \frac{\frac{1}{2} \cdot 4\pi r_{eff}^3 \cdot N_a}{3\eta_b \tau} \quad (2)$$

The effective volume $V_{eff}$ ($4\pi r_{eff}^3$) represents the volume in which the interaction can take place. This results in a half-sphere, since only half of the S1 molecules can interact with its corresponding receptor on the substrate.

**Culture of cell lines**. CHO cells were cultured in Ham's F12 medium (Sigma) supplemented with 10% FBS (Fetal Bovine Serum), penicillin (100 U mL$^{-1}$), streptomycin (100 μg mL$^{-1}$) (Invitrogen) and 2mM L-glutamine (Sigma). Lec2 cells were cultured in Mem α, nucleosides medium (Gibco) supplemented with 10% FBS,

penicillin (100 U mL$^{-1}$), streptomycin (100 μg mL$^{-1}$) (Invitrogen), and 2 mM L-glutamine (Sigma). HEK-293T cells were cultured in DMEM (Invitrogen) supplemented with 10% FBS, penicillin (100 U mL$^{-1}$), streptomycin (100 μg mL$^{-1}$) (Invitrogen). Cells were incubated at 37 °C with 5% of CO$_2$ and in an environment saturated in humidity.

**Transduction of Lec2 cells**. Lec2 cells were transduced to express nuclear eGFP as well as cytoplasmic mCherry using H2BeGFP and actin-mCherry-expressing lentiviruses as described previously[31].

**Labeling of UV-inactivated SARS-CoV-2 virions with Atto488 NHS ester dye**. In all, 200 μL of a $10^8$ particles mL$^{-1}$ solution of UV-inactivated SARS-CoV-2 virions were mixed with 10 μL of an Atto488 NHS ester dye (Atto-Tec) (10 mM in dry DMSO) under gentle agitation for 2 h in 500 μL of acetate buffer (pH 4.5). The free dyes were then eliminated and the fluorescently labeled virions concentrated to the initial concentration through filtration with an Amicon Ultra-0.5 centrifugal filter unit, MWCO 10 kDa (Sigma) (10 min centrifugation at 10 000 g for purification and 2 min at 1000 g for recovery of the product).

**Virus binding assay**. A co-culture of CHO and Lec2 (fluorescently labeled with actin-mCherry) was incubated for 1 h on ice (to prevent internalization) with a $10^8$ particles mL$^{-1}$ solution of UV-inactivated SARS-CoV-2 virions coupled to Atto488 NHS ester dye (see above). The cells were then rinsed three times with PBS and fixed with formaldehyde (4% in PBS for 15 min) (Invitrogen, Thermo Fisher Scientific). After a final wash with PBS, cells were imaged with laser scanning confocal microscopy (Zeiss LSM 980) using a 40x water objective. Images were analyzed with the Zen blue 2.3 software (Zeiss). A maximum intensity projection was performed to obtain a single image from the z-stack.

**FD-based AFM and fluorescence microscopy on living cells**. AFM correlative images of CHO and Lec2 cells were acquired using a Bioscope Resolve AFM (Bruker) in PeakForce QNM mode (Nanoscope software v9.2), which is coupled to an inverted epifluorescence microscope (Zeiss Observer Z.1) or confocal laser scanning microscope (Zeiss LSM 980)[31,32,35]. All the experiments were performed using a 40x oil objective (NA = 0.95). Cell images (30–50 μm$^2$) were recorded with forces of 500 pN using PFQNM-LC probes (Bruker) having tip lengths of 17 μm, tip radii of 65 nm and opening angles of 15°. All fluorescence and AFM experiments were realized under cell-culture conditions using the combined AFM and fluorescence microscopy chamber at 37 °C in either Mem α, nucleosides or Ham's F12 culture medium, depending on the cell type[35]. Cantilevers were calibrated using the thermal noise method[64], yielding values ranging from 0.08 to 0.14 N m$^{-1}$. The AFM tip was oscillated in a sinusoidal fashion at 0.25 kHz with a 750 nm amplitude. The sample was scanned using a frequency of 0.125 Hz and 128 or 256 pixels per line. AFM images and FD curves were analyzed using the Nanoscope analysis software (v1.9, Bruker), Origin, and ImageJ (v1.52e). Individual FD curves detecting unbinding events between the cell surface and S1 or SARS-CoV-2 were analyzed using the Nanoscope analysis and Origin software. The baseline of the retraction curve was corrected using a linear fit on the last 30% of the retraction curve. Using the force-time curve, the loading rate (slope) of each rupture event was determined. Optical images were analyzed using Zen Blue software (Zeiss)[32,35,37].

**Monitoring the effect of 9-AcSA porphyrin addition**. The live cell experiments were conducted in the same manner as described above by scanning a suitable area of confluent layers of cells, followed by adding either 10 nM, 100 nM, 1 μM or 10 μM of 9-AcSA porphyrin to the culture medium. The same area was then scanned again to monitor potential changes after addition of the 9-AcSA oligomer.

**Production of SARS-CoV-2 spike pseudotyped VSV virus**. The spike pseudotyped VSV virus were provided by Prof. Gert Zimmer (Switzerland) and produced as previously described[53]. Briefly, pCG1 SARS-CoV-2 spike with a C-terminal truncation of 18 amino acid residues plasmid was transfected in HEK-293T. The day after, VSV-deltaG virions were transduced in cells with MOI 5 per cell. After 1 hour of incubation at 37 °C with 5% of CO$_2$ and in an environment saturated in humidity, the media was removed and the cells were washed with PBS. The transduced cells were cultured in DMEM supplemented with 5% FBS, 1% penicillin, 1% streptomycin, 2mM L-Glutamine, 1 mM Na-Pyruvate, and NEAA, as well as VSV-G antibody (1:1000). The produced viruses were collected from the media the day after the transduction. Cell debris were cleared by centrifugation (1250 × g, 10 min) and with a 0.20 μm filter.

**Infectivity assay**. A549 or A549 cells stably overexpressing ACE2 ($1 \times 10^4$) were seeded in a 96-well plate[65]. The mixture of the pseudotyped virus at MOI 5 and either SA (Neu5Ac), 9-AcSA, SA-porphyrin or 9-AcSA-porphyrin at increasing concentration (0.001 μM, 0.01 μM, 0.1 μM, 1 μM, 10 μM) were incubated for 15 min at room temperature. The mixture was added in the media and the cells were incubated for 1 hour. The cells were washed with PBS and incubated in fresh cell culture media for 24 hours. The infectivity was monitored via fluorescence and

the images were taken with the bioimager device (Amersham Typhoon). The number of infected cells were counted by Fiji.

**Reporting summary**. Further information on research design is available in the Nature Research Reporting Summary linked to this article.

## Data availability

The datat that support this study are available from the corresponding auhtors upon reasonable request. The Source data underlying Figs. 2c, e, f, i; 3d, h, k; 5; and 6 are provided as a Source Data file.

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

## Acknowledgements

We thank you Gert Zimmer (Switzerland) for the VSV S-pseudotyped system. This work was supported by the Université catholique de Louvain, the Foundation Louvain and the Fonds National de la Recherche Scientifique (FRS-FNRS). This project received funding from the European Research Council under the European Union's Horizon 2020 research and innovation program (grant agreement no. 758224) and from the FNRS-Welbio (Grant # CR-2019S-01). The funders had no role in study design, data collection and analysis, decision to publish, or preparation of the manuscript. S.J.L.P., M.K., and D.A. are Research Fellow, postdoctoral researcher, and Research Associate at the FNRS, respectively. W.C. and S.P.V. are grateful to China Scholarship Council. R.J. received a post-doctoral fellowship from FNRS. S.B. was supported by the German Research Foundation (DFG) project numbers 415089553 (Heisenberg program), 240245660 (SFB1129), the state of Baden-Württemberg (AZ: 33.7533.-6-21/5/1), the Bundesministerium für Bildung und Forschung (BMBF) (01KI20198A) and within the Network University Medicine - Organo-Strat COVID-19. M.L.S. was supported by the BMBF (01KI20239B) and DFG project 416072091.

## Author contributions

S.P.V. and D.A. conceived the project. S.J.L.P., W.Z., M.K., R.J., B.J., D.M., and J.Y. planned the experiments and analyzed the data. S.J.L.P., M.K., D.M., and B.J. conducted the AFM experiments. W.C. and R.J. synthesized the SA derivatives. M.L.S. and S.B. produced the U.V. inactivated SARS-CoV-2 virions. J.Y. performed infection assays. All authors wrote the manuscript.

## Competing interests

S.P.V. and D.A. have applied for a patent for the use of 9-AcSA-glycoclusters to prevent SARS-CoV-2 infection (EP21195975). The remaining authors declare no competing interests.

## Additional information

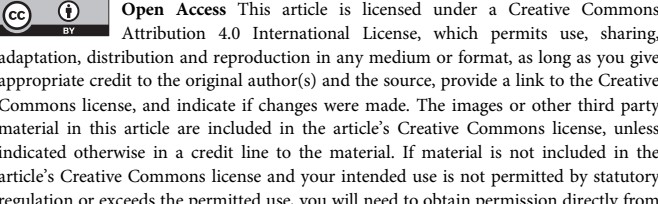

