## [Peer Review File · Nature Communications]

Multivalent 9-O-Acetylated-sialic acid glycoclusters as potent inhibitors for SARS-CoV-2 infectionReviewers' Comments:

Reviewer #1:

Remarks to the Author:

The authors apply a very novel method of FD-based AFM, which Alsteens pioneered amongst others, to quantify the early binding events of SARS-CoV-2 to sialic acids. The data which have been recorded with great care suggest that SARS-CoV-2 binds 9-O-acetylated-SA. The findings thus suggest that sialic acids can serve as a co-factor to promote the attachment of the virus to the host cell surface. Based on the insight revealed the authors have designed molecular compounds to block the binding of SARS-CoV-2. The blocking is facilitated by AcSA-derived glycoclusters, which suggests them as potent inhibitors to prevent SARS-CoV-infection. The manuscript is written very well, the experiments, the data analysis and the interpretation hereof are done with great care. I enjoyed reading the article as it in a very innovative way combines very promising technologies to characterize the early binding events of SARS-CoV-2 to sialic acids and to find ways to suppress this binding by the synthesis of novel SA derivatives/blocking molecules. I have a few questions which I would like to address to the authors. The authors may take them to further improve their excellent work.

The FD-based AFM experiments of the work are done with great care and represent state of the art. The results showing the specific interactions between SARS-CoV-2 virions and SA are exciting and unexpected. Although the results are convincing and controls have been with made with great care, they were all made using one relatively new method. One may thus think whether the author's work could benefit from an additional control, which applies a different independent method. As such the results could then be independently confirmed. As it is clear that there is probably no other method that can provide the quantitative parameters of the single interactions the authors provide about virus binding, the authors may think of using a qualitative method as independent control. For example one may test the specificity of the purified S1-glycoprotein or SARS-Cov-2 in binding to sialic acids, on both model surfaces and both living cell lines, by using time-lapse fluorescence microscopy. Particularly, combining fluorescence microscopy to the mixed two different cell lines as done in the FD-AFM experiments (Fig. 3) could prove to be quite convincing. However, this said it is clear that the outcome of the work would not change since the authors clearly demonstrated by the synthesis of SA derivatives/blocking molecules that virus binding can be successfully suppressed.

Minor issues

The second sentence of the abstract has some typo. One may simply replace 'other also' by 'others'.

The forth sentence of the abstract. The authors write that the 'SARS-CoV-2 binds specifically and preferentially with moderate affinity'. I have a problem understanding what is the difference between specifically and preferentially. Possibly there is some redundancy in the wording or the sentence may need some revision.

Introduction. The sentence 'In this work, we used force-distance (FD) curve-based atomic force microscopy (AFM),' would benefit from references to FD-based AFM as most of the readers will not know what this technology represents.

In Fig. 2 please explain at better detail what the individual dots represent and what the larger dots represent (averages?) Possibly in the legend after 'average forces' insert (large dots).

In Fig. 2i. pls explain what the numerals I-V stand for. I assume they stand for numeral rupture events, but this should be specified.

Fig. 3b and 3f. I would kindly suggest to also draw a dashed white line in the AFM height images so

one can better correlate height image and optical microscopy images.

Line 391. Please remove typo 'aggragation'

Reviewer #2:

Remarks to the Author:

The authors present an interesting manuscript on the interaction of the SARS CoV2 spike glycoprotein with sialic acid-containing glycans. The work is truly interdisciplinary, employing a variety of techniques, methods and protocols, which are well described, to monitor the binding and eventually to try to avoid the infection. For that, they employ diverse multivalent presentations of sialic acid derivatives, attached to a variety of chemical platforms. The results are promising.

The work is of significance to the field and the results globally support the conclusions.

There are some points that the authors should consider before publication.

1.-A recent publication, Moure et al. Angew Chem Int Ed Engl. 2022, Feb 22:e202201432. doi: 10.1002/anie.202201432 has seen that the spike recognises external sialic acids through its N-terminal domain. Although the binding, at the monomer level, is probably weak, it is specific. The manuscript probably appeared while the authors were submitting their manuscript. In any case, it should be cited and commented.

2.- The affinity is not spectacular in the AFM measurements (ca. 5 microM) for the 9-OAc derivative. In any case, it is better than for the non acetylated analogue and specific. However, for the in-cell experiments (CHO versus CHO-Lec2), there is not any mention to the role of the acetyl groups. Is the sialylation pattern known? The authors should comment on this point.

3.- The diverse functionalized scaffolds provide competition patterns that do not seem to properly correlate with the degree of functionalization. The presentation should be of paramount interest. Can the authors provide any perspective on that? Do the scaffolds aggregate themselves? Some sentences should be welcomed to explain the experimental observations.

Otherwise, it is a very nice paper, with interesting results and perspectives for the future.

Reviewer #3:

Remarks to the Author:

This manuscript uses atomic force microscopy to study early binding events of SARS-CoV-2 to the attachment factor 9-O-acetylated-sialic acid. The kinetics and affinity of this binding is discussed and various blocking molecules have been designed and used. This has led to the identification of cell binding and infection inhibitors. The Alsteens lab has previously published on force spectroscopy approaches to study virus-cell binding, including work on SARS-CoV-2 and the current manuscript is of the same high standard. There are quite a few things that need to be addressed, but overall I am very positive. Please see below my specific comments.

In the introduction it could be explained a bit better what 9-O-acetylated-SA (9-AcSA) is.

line 44, "are appearing with unprecedented speed". Why is this unprecedented? Because it never happened at all?

line 62, "or other mechanisms" this is a bit vague.

In fig 2b&2h the noise levels and number of data points per unit of length seems to vary quite a bit for

the different pulling speeds. Therefore the base lines look quite different for each situation. Wouldn't it make sense to have the same number of data points per unit length in order to compare the data better with each other?

In the legend of fig 2i it is not explained what the lines I-V signify and what the difference between the lines is.

The scale bar of the inset of fig 3b and 3f is missing. Also it is not very clear what is meant with adjacent cells and what the inset exactly is meant to show.

Line 288-289 says that the relative BF plots (Fig. 5c-f) show either no or slight inhibition for the SA-clusters. However, looking at the fullerene data it is clear that instead of no inhibition at 100 μM there is increased binding, without overlapping error bars with 0 μM . This is unexpected and a discussion on this point is missing.

Lines 425-431 The SARS-CoV2 imaging is performed in air, whereas, if I am not mistaken, all AFM data of the rest of the manuscript is obtained in liquid environments. Concluding that the particle integrity is maintained (line 155) from such an image of a dried virus is rather weak. So either remove this conclusion and the image, or perform the imaging in liquid (without having dried the particle in between) to make a good fit with the rest of the data in the manuscript.

Point-by-Point Response to the Reviewers Comments

Reviewer #1 (Remarks to the Author):

The authors apply a very novel method of FD-based AFM, which Alsteens pioneered amongst others, to quantify the early binding events of SARS-CoV-2 to sialic acids. The data which have been recorded with great care suggest that SARS-CoV-2 binds 9-O-acetylated-SA. The findings thus suggest that sialic acids can serve as a co-factor to promote the attachment of the virus to the host cell surface. Based on the insight revealed the authors have designed molecular compounds to block the binding of SARS-CoV-2. The blocking is facilitated by AcSA-derived glycoclusters, which suggests them as potent inhibitors to prevent SARS-CoV-infection. The manuscript is written very well, the experiments, the data analysis and the interpretation hereof are done with great care. I enjoyed reading the article as it in a very innovative way combines very promising technologies to characterize the early binding events of SARS-CoV-2 to sialic acids and to find ways to suppress this binding by the synthesis of novel SA derivatives/blocking molecules. I have a few questions which I would like to address to the authors. The authors may take them to further improve their excellent work.

Authors: The authors warmly thank the reviewer for his/her positive feedback and the kind words about our study. Below we have explained point-by-point how we addressed the remarks in the revised manuscript.

The FD-based AFM experiments of the work are done with great care and represent state of the art. The results showing the specific interactions between SARS-CoV-2 virions and SA are exciting and unexpected. Although the results are convincing and controls have been with made with great care, they were all made using one relatively new method. One may thus think whether the author's work could benefit from an additional control, which applies a different independent method. As such the results could then be independently confirmed. As it is clear that there is probably no other method that can provide the quantitative parameters of the single interactions the authors provide about virus binding, the authors may think of using a qualitative method as independent control. For example one may test the specificity of the purified S1-glycoprotein or SARS-Cov-2 in binding to sialic acids, on both model surfaces and both living cell lines, by using time-lapse fluorescence microscopy. Particularly, combining fluorescence microscopy to the mixed two different cell lines as done in the FD-AFM experiments (Fig. 3) could prove to be quite convincing. However, this said it is clear that the outcome of the work would not change since the authors clearly demonstrated by the synthesis of SA derivatives/blocking molecules that virus binding can be successfully suppressed.

Authors: As suggested by the reviewer, we performed an additional experiment with the aim to confirm our AFM results by an alternative qualitative method as an independent control. We assessed the binding between fluorescently labelled (using an Atto488 NHS ester dye) UV-inactivated SARS-CoV-2 virions and a co-culture of SA-expressing CHO (not fluorescently labelled) and non-SA-expressing Lec2 cells (fluorescently labelled with mCherry) using confocal microscopy. As shown in Figure R1 (now the SI Fig. 3 in the revised manuscript), most

of the virions (in green, highlighted in white circle) bind to the CHO and not to Lec2 cells (in red), supporting thus the role of SA in cellular adhesion and confirming our results.

Figure R1| Virus binding assay. Confocal microscopy of a coculture of SA-expressing CHO and fluorescently labelled Lec2 cells, deficient in SA expression (mCherry, red) incubated with UV-inactivated SARS-CoV-2 virions (fluorescently labelled with an Atto488 NHS ester dye, green and highlighted in white circle). Virions mainly bind to CHO cells, highlighting the role of SA in cellular adhesion.

Minor issues

1) The second sentence of the abstract has some typo. One may simply replace ‘other also’ by ‘others’.

Authors: We apologize for the typo. ‘Other also’ has now been replaced by ‘others’ in the revised manuscript.

2) The fourth sentence of the abstract. The authors write that the ‘SARS-CoV-2 binds specifically and preferentially with moderate affinity’. I have a problem understanding what is the difference between specifically and preferentially. Possibly there is some redundancy in the wording or the sentence may need some revision.

Authors: We clarified the sentence to avoid any confusion. The word “preferentially” has been deleted in the revised manuscript.

3) Introduction. The sentence ‘In this work, we used force-distance (FD) curve-based atomic force microscopy (AFM), ...’ would benefit from references to FD-based AFM as most of the readers will not know what this technology represents.

Authors: We thank the reviewer for raising this point. The revised manuscript includes now the citation of two recent reviews about FD-based AFM in the introduction (line 90).

[26] Viljoen, A. *et al.* Force spectroscopy of single cells using atomic force microscopy. *Nature Reviews Methods Primers* **1**, 63, doi:10.1038/s43586-021-00062-x (2021).

[27] Müller, D. J. *et al.* Atomic Force Microscopy-Based Force Spectroscopy and Multiparametric Imaging of Biomolecular and Cellular Systems. *Chem. Rev.*, doi:10.1021/acs.chemrev.0c00617 (2020).

4) In Fig. 2 please explain at better detail what the individual dots represent and what the larger dots represent (averages?) Possibly in the legend after 'average forces' insert (large dots).

Authors: We apologize for the lack of clarity. The legend of Fig. 2 and text (lines 131-132 and 153) have been adapted accordingly.

5) In Fig. 2i. pls explain what the numerals I-V stand for. I assume they stand for numeral rupture events, but this should be specified.

Authors: Sorry for the lack of clarity, that is now added in the legend. The numerals I-IV stand indeed for the single, double, triple and quadruple interaction, respectively.

6) Fig. 3b and 3f. I would kindly suggest to also draw a dashed white line in the AFM height images so one can better correlate height image and optical microscopy images.

Authors: We thank the reviewer for his/her relevant suggestion. A dashed white line was added in the optical microscopy images of Fig. 3b and 3f to better correlate with the AFM height and adhesion images.

7) Line 391. Please remove typo 'aggragation'

Authors: We are sorry for the typo that has been corrected in the revised manuscript.

Reviewer #2 (Remarks to the Author):

The authors present an interesting manuscript on the interaction of the SARS CoV2 spike glycoprotein with sialic acid-containing glycans. The work is truly interdisciplinary, employing a variety of techniques, methods and protocols, which are well described, to monitor the binding and eventually to try to avoid the infection. For that, they employ diverse multivalent presentations of sialic acid derivatives, attached to a variety of chemical platforms. The results are promising.

The work is of significance to the field and the results globally support the conclusions.

Authors: We are delighted to hear those positive comments about our work, thank you. Below we have explained point-by-point how we addressed the remarks in the revised manuscript.

There are some points that the authors should consider before publication.

1. A recent publication, Moure et al. *Angew Chem Int Ed Engl.* 2022, Feb 22:e202201432. doi: 10.1002/anie.202201432 has seen that the spike recognises external sialic acids through its N-terminal domain. Although the binding, at the monomer level, is probably weak, it is specific. The manuscript probably appeared while the authors were submitting their manuscript. In any case, it should be cited and commented.

Authors: We thank the reviewer for pointing out this publication, that appeared indeed during the submission of our manuscript. We added a sentence in the introduction (lines 70-72) to comment that the reference supplies a first experimental demonstration of the existence of a SA binding site in the N-terminal domain (NTD) of the S1 protein. However, we would like to indicate that the SA residues studied in the mentioned paper are α 2,3 and α 2,6 sialyl *N*-acetylglucosamine, and not 9-AcSA as in our own work. The latter has been highlighted in the updated Fig. 1a.

2. The affinity is not spectacular in the AFM measurements (ca. 5 μ M) for the 9-OAc derivative. In any case, it is better than for the non acetylated analogue and specific. However, for the in-cell experiments (CHO versus CHO-Lec2), there is not any mention to the role of the acetyl groups. Is the sialylation pattern known? The authors should comment on this point.

Authors: We thank the reviewer for the relevant comment that was indeed not clear enough in our manuscript. Although the sialylation pattern of the CHO cell line is not fully elucidated, the cells have been shown to support the infection by bovine coronavirus that uses 9-AcSA as an entry receptor, as shown by Berting et al. (2010), ref [40] in the manuscript. Additionally, our experimental results support the expression of 9-AcSA on the CHO cell surface, as the data are in good agreement with the data collected on model surfaces. The revised manuscript includes now a more detailed explanation about this matter (line 165).

More strikingly, we chose CHO derived cell lines, because they possess interesting mutations, such as the Lec2 cell line. This cell line is 70-90% deficient of SA in their glycoproteins and gangliosides, thus very suitable as a control for our experiments. Please see ref [41] in the manuscript.

Finally, from a broader point of view, the fact that the cell line could possess different sialylation patterns is an asset as we can show, that our molecule based on 9-AcSA (and

particularly the porphyrin) stably binds to the virus, reducing both the binding and the infectivity.

- 40 Berting, A., Farcet, M. R. & Kreil, T. R. Virus susceptibility of Chinese hamster ovary (CHO) cells and detection of viral contaminations by adventitious agent testing. *Biotechnol Bioeng* **106**, 598-607, doi:10.1002/bit.22723 (2010).
- 41 Deutscher, S. L., Nuwayhid, N., Stanley, P., Briles, E. I. B. & Hirschberg, C. B. Translocation across Golgi vesicle membranes: a CHO glycosylation mutant deficient in CMP-sialic acid transport. *Cell* **39**, 295-299 (1984).

3. The diverse functionalized scaffolds provide competition patterns that do not seem to properly correlate with the degree of functionalization. The presentation should be of paramount interest. Can the authors provide any perspective on that? Do the scaffolds aggregate themselves? Some sentences should be welcomed to explain the experimental observations.

Authors: Thank you for the interesting remark. Indeed, we do not observe a correlation between the degree of functionalization and blocking efficiency. This can have diverse explanations (accessibility of the 9-AcSA residues, the flexibility of the scaffolds, the distance between the residues to bind different S1 simultaneously, 3D conformation of the S1, ...) that are all difficult to predict theoretically. This is why we tested an array of scaffolds with various topologies and valencies before characterizing more in depth the most promising compound. In addition, we are discussing in lines 315-323 the aggregation behavior of porphyrins that should explain its remarkable efficacy at low concentrations.

Otherwise, it is a very nice paper, with interesting results and perspectives for the future.

Authors: We thank the reviewer for the interest he/she brought to our study and the suggestions to improve our manuscript!

Reviewer #3 (Remarks to the Author):

This manuscript uses atomic force microscopy to study early binding events of SARS-CoV-2 to the attachment factor 9-O-acetylated-sialic acid. The kinetics and affinity of this binding is discussed and various blocking molecules have been designed and used. This has led to the identification of cell binding and infection inhibitors. The Alsteens lab has previously published on force spectroscopy approaches to study virus-cell binding, including work on SARS-CoV-2 and the current manuscript is of the same high standard. There are quite a few things that need to be addressed, but overall I am very positive. Please see below my specific comments.

Authors: We thank the reviewer for his/her positive comments that will enhance the quality of our work. Below we have explained point-by-point how we addressed the remarks in the revised manuscript.

In the introduction it could be explained a bit better what 9-O-acetylated-SA (9-AcSA) is.

Authors: We apologize for the lack of explanation about 9-O-acetylated-SA. Figure 1 has now been adapted to better illustrate the diversity of the SA family and the chemical structure of 9-AcSA.

line 44, "are appearing with unprecedented speed". Why is this unprecedented? Because it never happened at all?

Authors: We agree that we cannot really judge the "unprecedented" nature of the situation. The sentence is now corrected in the revised manuscript by deleting "with unprecedented speed".

line 62, "or other mechanisms" this is a bit vague.

Authors: Thank you for your remark, we now removed this part of the sentence.

In fig 2b&2h the noise levels and number of data points per unit of length seems to vary quite a bit for the different pulling speeds. Therefore the base lines look quite different for each situation. Wouldn't it make sense to have the same number of data points per unit length in order to compare the data better with each other?

Authors: We thank the reviewer for raising this concern, that is in fact intrinsic to FD curves recording by AFM. The data shown in Fig. 2b and 2h present the raw data collected by AFM

using single-molecule force spectroscopy. As these experiments were performed at various speeds, the signal processing board collects the data at a fixed frequency resulting in fewer data points processed by the computer for the highest speeds. In turn, the force-distance curves appeared to have fewer data points per unit of length. In the data collection process, only the number of data point per unit of time is conserved.

In the legend of fig 2i it is not explained what the lines I-V signify and what the difference between the lines is.

Authors: We apologize for the missing information, that is now added in the legend. The numerals I-IV stand indeed for the single, double, triple and quadruple interaction, respectively.

The scale bar of the inset of fig 3b and 3f is missing. Also it is not very clear what is meant with adjacent cells and what the inset exactly is meant to show.

Authors: Thank you for the comment that will allow us to better explain our experimental setup. We are working with a co-culture of CHO and Lec2 cells, meaning that they are grown as a mixed culture (a co-culture) in the same petri dish, used for our AFM experiments. As our goal is to probe the two cell lines simultaneously in one image to have a direct, “real-time” control for specific interaction, we need to find areas where those two cell lines (CHO and Lec2) are right next to each other (“adjacent”). To be able to distinguish between the two cell lines, we use fluorescently labelled Lec2 cells. Guided by fluorescence, the insert represents the optical image at the position probed by AFM. We have now added the scale bar of the insets presented in Fig. 3b and Fig. 3f and clarified the legend of the figure. We apologize for the missing information.

Line 288-289 says that the relative BF plots (Fig. 5c-f) show either no or slight inhibition for the SA-clusters. However, looking at the fullerene data it is clear that instead of no inhibition at 100 μM there is increased binding, without overlapping error bars with 0 μM . This is unexpected and a discussion on this point is missing.

Authors: Thank you for the comment. At first sight, we were also wondering why the binding probability is increasing with 100 μM of SA-fullerene. However, the statistical analysis (two sample t-test) reveals that this difference is not significant. We thus cannot conclude and discuss a significant increase in binding.

Lines 425-431 The SARS-CoV2 imaging is performed in air, whereas, if I am not mistaken, all AFM data of the rest of the manuscript is obtained in liquid environments. Concluding that the particle integrity is maintained (line 155) from such an image of a dried virus is rather weak. So either remove this conclusion and the image, or perform the imaging in liquid (without having dried the particle in between) to make a good fit with the rest of the data in the manuscript.

Authors: The reviewer is right about the mismatch between the AFM (force spectroscopy) data acquired in liquid and the topography imaging of the virus performed in air. Nevertheless, we still consider the control of the virion structure integrity as successful because: (i) we obtain similar results to other groups (see below) and (ii) the drying would add more stress

that could potentially destabilize the virions rather than the other way round. We thus show that UV radiation and drying does not affect virion integrity.

Lyonnais, S., Hénaut, M., Neyret, A., Merida, P., Cazevielle, C., Gros, N., ... & Muriaux, D. (2021). Atomic force microscopy analysis of native infectious and inactivated SARS-CoV-2 virions. *Scientific reports*, 11(1), 1-7.

Kiss, B., Kis, Z., Pályi, B., & Keller Mayer, M. S. (2021). Topography, spike dynamics, and nanomechanics of individual native SARS-CoV-2 virions. *Nano letters*, 21(6), 2675-2680.